# Investigating the Effect of Bacilli and Lactic Acid Bacteria on Water Quality, Growth, Survival, Immune Response, and Intestinal Microbiota of Cultured *Litopenaeus vannamei*

**DOI:** 10.3390/ani14182676

**Published:** 2024-09-14

**Authors:** Ana Sofía Vega-Carranza, Ruth Escamilla-Montes, Jesús Arturo Fierro-Coronado, Genaro Diarte-Plata, Xianwu Guo, Cipriano García-Gutiérrez, Antonio Luna-González

**Affiliations:** 1Instituto Politécnico Nacional, Centro Interdisciplinario de Investigación para el Desarrollo Integral Regional-Unidad Sinaloa, Departamento de Acuacultura, Boulevard Juan de Dios Bátiz Paredes #250, Col. San Joachín, Guasave 81101, Sinaloa, Mexico; asvc_1394@hotmail.com (A.S.V.-C.); arturofierrojr@hotmail.com (J.A.F.-C.); gdiarte@ipn.mx (G.D.-P.); cgarciag@ipn.mx (C.G.-G.); 2Instituto Politécnico Nacional, Centro de Biotecnología Genómica, Boulevard del Maestro S/N Esquina Elías Piña, Col. Narciso Mendoza, Reynosa 88710, Tamaulipas, Mexico; xguo@ipn.mx

**Keywords:** nitrifying bacteria, *Litopenaeus vannamei*, intestinal microbiota, water quality

## Abstract

**Simple Summary:**

In hyperintensive culture systems, high stocking density produces excess nitrogenous wastes, which deteriorate water quality, affecting the physiology and gut microbiota of shrimp. In this study, different methodologies were used to analyze water quality, immune systems and intestinal microbiota. The results showed that bacilli and LAB, inoculated into water, utilize nitrogen in several ways, improving shrimp growth and survival. In addition, the bacteria modulate the intestinal microbiota and immune response, improving shrimp production. This is the first time that the use of *Pediococcus pentosaceus* and *Leuconostoc mesenteroides* as nitrifying bacteria has been reported.

**Abstract:**

Shrimp is one of the most important aquaculture industries. Therefore, we determined the effect of nitrifying-probiotic bacteria on water quality, growth, survival, immune response, and intestinal microbiota of *Litopenaeus vannamei* cultured without water exchange. In vitro, only *Bacillus licheniformis* used total ammonia nitrogen (TAN), nitrites, and nitrates since nitrogen bubbles were produced. TAN decreased significantly in the treatments with *B. licheniformis* and *Pediococcus pentosaceus* and *Leuconostoc mesenteroides*, but no differences were observed in nitrites. Nitrates were significantly higher in the treatments with bacteria. The final weight was higher only with bacilli and bacilli and LAB treatments. The survival of shrimp in the bacterial treatments increased significantly, and superoxide anion increased significantly only in lactic acid bacteria (LAB) treatment. The activity of phenoloxidase decreased significantly in the treatments with bacteria compared to the control. Shrimp treated with bacilli in the water showed lower species richness. The gut bacterial community after treatments was significantly different from that of the control. Linoleic acid metabolism was positively correlated with final weight and superoxide anion, whereas quorum sensing was correlated with survival. Thus, bacilli and LAB in the water of hyperintensive culture systems act as heterotrophic nitrifers, modulate the intestinal microbiota and immune response, and improve the growth and survival of shrimp. This is the first report on *P. pentosaceus* and *L. mesenteroides* identified as nitrifying bacteria.

## 1. Introduction

From an economic and nutritional point of view, shrimp is the most important aquaculture market, and their production is over 4 million tons annually [1]. Shrimp hyperintensive culture is a culture system conducted in open-air ponds or ponds covered with a plastic membrane ranging from 0.5 to 1 ha and is characterized by stocking densities exceeding 100 organisms m^−3^. Therefore, it is necessary to supplement oxygen to the water through aeration equipment, which allows for improved culture conditions and optimized feeding [2]. Due to the high stocking density, excess nitrogenous waste (ammonium and nitrites) is generated, which deteriorates the water quality and makes it toxic to cultured organisms. The main ammonium sources are shrimp excretions and sediment derived from mineralization of organic matter and molecular diffusion of reduced sediment [3,4].

Nitrification is the general biochemical process of oxidation of ammonium (NH_4_^+^) to nitrite (NO_2_^−^) and, finally, to nitrate (NO_3_^−^) [5]. Bacterial nitrification is one of the commonly used methods for ammonium elimination from aquaculture systems without water exchange [6]. The oxidation of ammonium to nitrite is generally conducted by bacteria that possess the ammonia–oxygenase enzyme such as *Nitrosomonas*, *Nitrosococcus*, *Nitrosospira*, *Nitrosolobus*, and *Nitrosovibrio*, whereas oxidation of nitrite to nitrate can be produced by the genera *Nitrospina*, *Nitrococcus,* and *Nitrobacter* using the nitrite–oxidase enzyme. However, these genera of anaerobic bacteria are more vulnerable to changes in the environment and have a slower metabolism [7,8]. Therefore, there are other bacterial groups used for bioremediation in aquaculture, which include *Bacillus licheniformis*, *Bacillus subtilis*, *Bacillus cereus*, *Pseudomonas*, and *Paracoccus* species [9,10,11]. These bacteria are also recognized as probiotics in aquaculture because they provide protection against bacterial and viral invasions in shrimp by stimulating the immune response (cellular and humoral reactions) [12,13].

The invertebrate immune system is made up of cellular (hemocytes) and humoral effectors. Hemocytes, which are the first line of defense, participate in phagocytosis, capsule and nodule formation, cell adhesion, cytotoxicity, and coagulation [14]. Humoral effectors found in plasma include antimicrobial peptides, lysozyme, lysosomal hydrolytic enzymes, lectins, the prophenoloxidase system, α2-macroglobulin, and transglutaminase [15,16,17,18].

Some bacterial species are culturable in the bacteriological media used in aquaculture. However, they could not offer an accurate picture as other species of bacteria could be present but hardly, or even not, culturable [17]. Therefore, metagenomics can be used to know the structure and function of the microorganisms of a specific sample [18]. The modulation of the intestinal microbiota in cultured shrimp has been conducted with the addition of probiotic bacteria [19]. The study of the microbial communities in the digestive tract of aquatic animals aims to highlight the benefit provided by the microbe–host relationship, which influences the health of the organism and its protection against pathogens [20].

Therefore, in this work, the effect of nitrifying-probiotic bacteria on water quality, growth, intestinal microbiota, immune response, and survival of *Litopenaeus vannamei* cultured with zero water exchange was determined. This study will allow us to better understand the effect of heterotrophic bacteria such as bacilli and LAB on water quality and shrimp physiology, as well as gut microbiota.

## 2. Materials and Methods

### 2.1. Bacillus Licheniformis BCR 4-3 and Lactic Acid Bacteria (LAB) Culture

*Bacillus licheniformis* BCR 4-3, isolated and characterized by Escamilla–Montes et al. [21] from shrimp intestine, was grown in trypticase soy broth (BD Bioxon^®^, Cuautitlán Izcalli, Estado de Mexico, Mexico) with 2.5% NaCl at 32 °C for 24 h and centrifuged at 5000× *g* for 20 min. Lactic acid bacteria (LAB), such as *Pediococcus pentosaceus*, isolated and characterized by Leyva–Madrigal et al. [22] from shrimp intestine, and *Leuconostoc mesenteroides*, isolated and characterized by Escamilla–Montes et al. [21] from shrimp intestine, were grown in Man Rogosa and Sharpe (MRS, BD Difco^®^, Cuautitlán Izcalli, Estado de Mexico, Mexico) broth with 2.5% NaCl at 32 °C for 48 h and centrifuged at 5000× *g* for 20 min.

The bacilli and LAB pellets were resuspended in a 2.5% NaCl solution and brought to an absorbance (580 nm) of 1.0 in a Thermo Spectronic Genesys 2^®^ spectrophotometer (Thermo Fisher Scientific, Inc., Waltham, MA, USA) before being inoculated into the shrimp culture water according to Escamilla–Montes et al. [21], who counted the CFU. 

### 2.2. In Vitro Utilization of Ammonium, Nitrites, and Nitrates by Bacilli and LAB 

The medium used for the study of nitrification and denitrification consists of 20 g of peptone, 2 g of (NH_4_)_2_SO_4_, 1.5 g of KNO_3_, 1.5 g of MgSO_4_, 1.5 g of KH_2_PO_4_, and 20 g of NaCl per 1000 mL of distilled water. The medium was distributed, in triplicate, in test tubes with 15 mL each, to which a Durham tube was placed and inoculated with the isolated strains, leaving three tubes uninoculated as a control. They were incubated at 32 °C, and positive or negative gas (nitrogen) formation was observed in the Durham tubes for 1 week.

### 2.3. Experimental Design

#### 2.3.1. Origin and Acclimatization of Experimental Shrimp

Experimental shrimp (0.39 ± 0.09 g) were obtained from the hatchery systems of the Acuícola Cuate Machado farm (Guasave, Sinaloa, Mexico). Animals were transported to the Aquaculture Laboratory at CIIDIR-Sinaloa in 250 L plastic containers with 200 L of seawater from the culture tanks and constant aeration. Hypersaline water (90–100 PSU), previously treated with liquid chlorine (1.5 mL L^−1^) for 1 day, was used. After chlorination, water was aerated with diffuser stones to eliminate the chlorine by volatilization. The shrimp were acclimatized for 15 d at 30 PSU in 1000 L plastic tank with 300 L of seawater, kept at room temperature and constant aeration [23], and fed with commercial feed (35% Purina^®^, Miguel Hidalgo, Ciudad de Mexico, Mexico) at 08:00, 13:00, and 17:00 h.

#### 2.3.2. Bioassay: Effect of Bacilli and LAB on Water Quality, Growth, Survival, Immune System, and Intestinal Microbiota of Shrimp

The bioassay lasted 35 d and began with shrimp weighing 0.74 ± 0.07 g. Shrimp were cultured in plastic tanks (30 L) with 20 L of filtered seawater (20 µm), 30 PSU, and constant aeration. Twelve animals were placed per tank (equivalent to 600 shrimp m^3^) and fed three times a day (08:00, 13:00, and 16:00 00 h) with commercial feed (35% Purina^®^, Mexico), adjusting the amount of feed according to the shrimp biomass (weight). The bioassay consisted of four treatments, each treatment with three replicates: (I) Control without bacteria in the water; (II) bacilli in the water; (III) LAB in the water; and (IV) bacilli + LAB in the water. Each bacterium (3 × 10^6^ CFU·L^−1^) was placed in the water every 7 d. There was no cleaning of the tanks or water exchange during the experimental period. The determination of physicochemical parameters (dissolved oxygen, temperature, pH, and salinity) and survival were determined daily. Total ammonia nitrogen (TAN = NH_3_ and NH_4_^+^), nitrites, and nitrates were determined on Days 7, 15, and 30 after the start of the bioassay.

The photoperiod was a 12:12 h light: dark cycle. Values of temperature (T), pH (HI 98127 pHep, Hanna Instruments, Woonsocket, RI, USA), salinity (S, Refractometer W/ATC 300011, Sper Scientific, Scottsdale, AZ, USA), and dissolved oxygen (YSI model 55 oximeter, Yellow Spring Instruments, Yellow Springs, OH, USA) were determined daily. During the bioassay, the physicochemical parameters behaved as shown in Table 1, indicating an optimal range [24].

On the 35th day, the shrimp were weighed, and five shrimp were taken from the three tanks of each treatment (2:2:1) to obtain intestine samples. The intestine of each shrimp was dissected, placed in a 1.5-mL Eppendorf tube with 1 mL of 96% (*v*/*v*) ethanol, and stored at −80 °C. The samples (five per treatment) were sent to the Research Center for Food and Development (CIAD, Mazatlán, Sinaloa, Mexico) for bacterial DNA extraction, library preparation, and sequencing in Illumina MiniSeq.

Initial and final weight were determined for each treatment. Also, specific growth rate (SGR, %d^−1^) was calculated as follows [25]:$$SGR(\%d^{-1})=\frac{100(lnW2-lnW1)}{t2-t1}$$
where W_1_ and W_2_ are the weights of the shrimp at times t_1_ and t_2._

#### 2.3.3. Hemolymph Collection

Nine shrimp per treatment were obtained for the individual extraction of hemolymph (3:3:3). Hemolymph was withdrawn from intermolt shrimp in the ventral sinus (second pair of pleopods) using a 1 mL tuberculin syringe with a 25-gauge needle. The syringe was previously loaded with precooled anticoagulant (450 mM NaCl, 10 mM KCl, 10 mM Hepes + 10 mM EDTA-Na_2_, pH 7.3) [26] in a ratio of 3:1 (three volumes of anticoagulant for each volume of hemolymph).

#### 2.3.4. Hemocyte Count

Fifty microliters of anticoagulant-hemolymph from the nine shrimp were diluted in 150 µL of formaldehyde (6%), and then 10 µL were placed in a hemocytometer (Neubauer chamber) to count hemocytes (THC) using a compound microscope (Labomed, Labo America, Inc., Fremont, CA, USA).

#### 2.3.5. Superoxide Anion

The free radical superoxide anion was quantified according to Song and Hsieh [27]. Samples (100 μL) of anticoagulant–hemolymph from three shrimp per tank (nine per treatment) were centrifuged at 800× *g* for 10 min at 4 °C, and the plasma was discarded. Hemocyte pellets from each shrimp were washed three times with SIC-EDTA buffer and stained with 100 μL of nitro-blue tetrazolium (NBT) solution (0.3%) for 30 min at 37 °C. The superoxide anion reduces NBT to a formazan. The reaction was completed through the elimination of the NBT solution by centrifuging and adding 100 μL of absolute methanol to the cell pellet. After three washings with 70% methanol, hemocytes were air-dried for 30 min, and 140 μL of DMSO and 120 μL KOH (2 M) were added to dissolve the cytoplasmic formazan. The samples were centrifuged, and the supernatant was placed in new tubes. The absorbance of the dissolved formazan was read at the optical density (OD) at 630 nm in a Thermo Spectronic Genesys 2 Spectrophotometer 2.10 (Thermo Fisher Scientific).

#### 2.3.6. Phenoloxidase Activity (PO) in Hemolymph

Hemolymph from three shrimp per tank (nine shrimp per treatment) was subjected to freeze–thaw cycle (−20 °C). Finally, the samples were centrifuged at 14,000× *g* for 10 min at 4 °C to obtain the hemolymph supernatant. As the extraction buffer containing EDTA, which chelates calcium in plasma that is necessary for activating the prophenoloxidase activating enzyme, prophenoloxidase was activated with trypsin. Fifty microliters of sample were incubated with 50 μL of trypsin (0.1 mg mL^−1^) and incubated for 30 min at 37 °C. Next, the sample was incubated with 50 μL of L-DOPA (3 mg mL^−1^ of distilled water) for 10 min at 37 °C. Phenoloxidase activity was measured spectrophotometrically by recording the formation of dopachrome produced from L-DOPA at 492 nm in a microplate reader spectrophotometer from Awareness Technology (Palm City, FL, USA). The color developed was measured against a blank with L-DOPA [26]. One unit of PO activity was defined as an absorbance increase of 0.001/min/mg protein. Protein content in samples was determined by the Bradford method [28] using bovine serum albumin (protein) as a standard.

### 2.4. Water Sampling, TAN, Nitrites, and Nitrates Determination

Samples of 1000 mL of culture water were taken from each tank and placed in plastic bottles with screw caps (1000 mL). The water that was taken was replaced with clean water. The samples were shaken, and the solids were allowed to settle down for 3 min. Subsequently, samples were filtered by gravity with 696-grade fiberglass filters of 4.7 cm diameter (VWR International, Lutterworth, Leicestershire, UK). The concentration of TAN, nitrites, and nitrates was determined following the Practical Manual of Seawater Analysis [29].

### 2.5. Metagenomic Analysis

#### 2.5.1. Extraction of Bacterial DNA, Library Preparation, and Sequencing in Illumina MiniSeq

Microbial DNA was extracted from intestine samples using the cetyltrimethylammonium bromide (CTAB) method [30]. The variable region V3 of the bacterial 16S rRNA gene was amplified by PCR with the primers 338F (ACT CCT ACG GGAGGC AGC AG) and 533R (TTA CCG CGG CTG CTG GCAC) [31]. DNA amplification was conducted with the KAPA kit (2x KAPA HiFi HotStart ReadyMix) from Roche (Basel, Switzerland) in a 25 μL reaction volume. PCR was performed in a thermal cycler using the following program [32]: one cycle of 30 s at 95 °C, followed by 25 cycles, each one of 30 s at 95 °C, 55 °C for 30 s, 72 °C for 15 s, and a final extension at 72 °C for 7 min. AMPure XP magnetic beads were used to clean up amplicons from free primers and primer dimers. For sequencing, purified amplicons were associated with dual indices and Illumina sequencing adapters using the Nextera XT index kit (Illumina, San Diego, CA, USA). Illumina MiniSeq platform was used under standard conditions (300 cycles, 2 × 150 pair-end) to perform sequencing. Before their quantification, the libraries were purified with AMPure XP magnetic beads. Raw reads from Illumina MiniSeq sequencing were deposited in the NCBI Sequence Read Archive (SRA) under accession number PRJNA1044443.

#### 2.5.2. Gut Bacterial Taxonomy, Abundance, and Diversity Analysis

The raw sequences were cleaned with pair-end cleaner v. 1.0.3 and then analyzed with the web-based Shaman (Paris, Seine, France) [33] and MicrobiomeAnalyst (Ste. Anne de Bellevue, Quebec, Canada) [34,35], platforms for microbial taxonomy, abundance, and diversity. The analysis of read-quality control, dereplication, removing singletons, removing chimera sequences, and grouping was conducted on the Shaman platform to construct operational taxonomic units (OTU). The OTUs shared by the three groups were determined using the Venn diagram analysis [36,37]. On the Shaman platform, the reads obtained from the V3 hypervariable region of the bacterial 16S rRNA gene were annotated against the SILVA version 138.1 (Paris, Seine, France) [38] ribosomal RNA database with a confidence threshold of 0.8. The analyses of the alpha (Shannon, Simpson, Chao 1, ACE) and beta [Non-metric multidimensional scaling (NMDS)] indices were performed in the MicrobiomeAnalyst platform to explore the effects of bacilli and LAB in water on the bacterial community composition of cultured shrimp intestines.

Bacterial functional profile of the white shrimp intestine was performed on the multimodular web platform iVikodak (Pune, Maharastra, India). Using the Global Mapper module (independent contribution algorithm), functional profiles were inferred based on the KEGG database pathways [39].

### 2.6. Statistical Analysis

Most data are shown as mean ± SD. One-way analysis of variance (ANOVA) was applied to determine the differences in concentration of ammonium, nitrites, nitrates, SGR, survival, total hemocytes, PO activity, and superoxide anion. Survival data in percentage were arcsine-transformed according to Daniel [40]. If significant differences were found in the ANOVA, a Tukey’s HSD test was used to identify these differences at *p* < 0.05. For alpha diversity (Shannon, Simpson, Chao 1, ACE), the Kruskal–Wallis test was used (*p* < 0.05). For beta diversity analysis, the ANOSIM test (*p* < 0.05) was performed in the MicrobiomeAnalyst web-based platform. Spearman correlation analysis was performed to correlate some functional profiles with immune and productive variables (*p* < 0.05). Correlation plots were made online with SRplot (Changsha, Hunan, China) [41].

## 3. Results

### 3.1. In Vitro Utilization of Ammonium, Nitrites, and Nitrates by Bacilli and LAB

Results showed that during in vitro tests, bacilli oxidizes TAN and nitrites (nitrification process) and reduces nitrates (denitrification process) since nitrogen bubbles were produced from the chemical substances provided in the culture medium. Each LAB grew in the culture medium but did not produce nitrogen bubbles.

### 3.2. Effect of Bacilli and LAB on Culture Water Quality

Unlike nitrites and nitrates, the TAN concentration in the control group, on Days 7 and 15, showed a trend to increase, but no significant differences were observed with respect to the treatments with bacteria. On the 30th day, the TAN concentration of water in the control (I), bacilli in the water (II), LAB in the water (III), and bacilli and LAB in the water (IV) was 0.3 ± 0.029, 0.030 ± 0.001, 0.034 ± 0.004, and 0.015 ± 0.001 mg L^−1^, respectively. A significant decrease (*p* < 0.05) was observed in Treatments II, III, and IV with respect to the control (I). No significant differences in the concentration of nitrites (0.40–0.46 mg L^−1^) were observed among Treatments (*p* > 0.05). The nitrate concentration of water in the control (I), bacilli in the water (II), LAB in the water (III), and bacilli and LAB in the water (IV) was 2.47 ± 0.08, 3.27 ± 0.16, 3.44 ± 0.13, and 4.10 ± 0.02 mg L^−1^, respectively. An increase (*p* < 0.05) was observed in Treatments II, III, and IV with respect to the control (I). Treatment IV presented a significant increase (*p* < 0.05) compared to Treatments II and III (Figure 1).

#### 3.2.1. Shrimp Growth

The initial average weight was between 0.70 g (control) and 0.73–0.79 g (Treatments), no significant differences were observed between the Treatments with bacteria in the water and the control (*p* > 0.05). The average final weight was between 3.09 g (control) and 3.87–4.18 g (Treatments with bacteria in the water). Significant differences were observed between the control and bacilli in the water (II), and between the control and bacilli and LAB in the water (IV) (*p* < 0.05). The SGR was between 4.16 ± 0.25% d^−1^ and 4.78 ± 0.21% d^−1^. No significant differences were observed between the Treatments and the control (*p* > 0.05) (Table 2).

#### 3.2.2. Shrimp Survival

On the 34th day, dead shrimp were found in the tanks. The survival of shrimp in the control group (I), bacilli in the water (II), LAB (III), and bacilli and LAB in the water (IV) on the 35th day was 60 ± 12.72%, 100 ± 0.0%, 95 ± 4.81%, and 95 ± 4.81%, respectively. Significant differences were found between control and Treatments (*p* < 0.05) (Figure 2).

### 3.3. Immune Parameters

#### 3.3.1. Total Hemocyte Count

Figure 3 shows the results of the number of shrimp hemocytes in the control and treatment groups with bacteria in water (II, III, IV). The number of hemocytes ranged between 15.38 (×10^6^) and 18.60 (×10^6^) cells mL^−1^. No significant differences were observed (*p* > 0.05).

#### 3.3.2. Superoxide Anion

Figure 4 shows the results of superoxide anion concentration (absorbance at 630 nm) in shrimp hemocytes. A significant increase (*p* < 0.05) was observed in Treatment III (LAB, 0.418 ± 0.062) compared to I (control, 0.275 ± 0.057), II (bacilli in the water, 0.299 ± 0.019), and IV (bacilli and LAB in the water, 0.277 ± 0.054).

#### 3.3.3. Phenoloxidase Activity in Shrimp Hemolymph

Phenoloxidase activity was detected in shrimp hemolymph (Figure 5). The phenoloxidase activities in the hemolymph of the control group (I), bacilli in the water (II), LAB in the water (III), and bacilli and LAB in the water (IV) were 0.60 ± 0.13, 0.34 ± 0.06, 0.45 ± 0.12, and 0.44 ± 0.11 units, respectively. A significant decrease (*p* < 0.05) of enzyme activity was observed in Treatments compared to the control.

### 3.4. Gut Microbiota Analysis

In total, 428 OTUs were annotated by the Silva ribosomal RNA database. The control had the highest number (391) of OTUs and the Treatment with bacilli in the water had the lowest (278). According to the Venn analysis, the four groups shared 189 OTUs. Regarding the unique OTUs, the group with the most OTUs was the control (26) and the one with the least was the mixture of bacilli and LAB in the water (6) (Figure 6).

#### 3.4.1. Relative Bacterial Abundance in the Shrimp Gut

The relative abundance of the most relevant taxa at the phylum and genus levels in the intestines of shrimp treated with bacteria (bacilli and LAB) in the water and control was determined. The most abundant taxa (Figure 7) in control and Treatments were proteobacteria and bacteroidota, and no significant differences were observed among the treatment groups with bacteria in the water (*p* > 0.05). Patescibacteria, the third phylum in relative abundance, was significantly enriched (*p* < 0.05) in Treatment II (bacilli in the water) compared to I (control), III (LAB in the water), and IV (bacilli and LAB in the water).

Results show the most abundant genera composed of *Vibrio*, *Pseudoalteromonas*, *Mesonia*, *Gilvimarinus*, *Staphylococcus*, *Tenacibaculum*, *Alkalimarinus*, *Yoonia-Loktanella*, *Salinimicrobium*, and *Halioglobus.* Among these 10 genera, *Vibrio* was the most abundant, followed by *Pseudoalteromonas.* Only *Pseudoalteromonas* decreased significantly (*p* < 0.05) in Treatments II (bacilli in the water) and III (LAB in the water) as compared to control and Treatment IV (bacilli and LAB in the water) (Figure 8).

#### 3.4.2. Alpha Diversity Indices

Good’s estimated sample coverage in the three groups was 99.93–99.99%, which means adequate data sampling and good-quality sequences. Rarefaction analysis showed that samples reached the plateau, which means that the sequencing depth was sufficient to capture all the OTUs in the samples. No significant differences were found in Shannon and Simpson indices. However, they showed a tendency to be lower in the treatment groups with bacteria in the water. In the Chao index, the values were between 93.95 ± 25.25 and 149.74 ± 27.71; significant differences were observed between Treatment II and control, Treatment III, and Treatment IV (*p* < 0.05). In the ACE, the values were between 88.95 ± 28.97 and 146.34 ± 27.87, indicating that there were significant differences between the control and Treatment II (*p* < 0.05) (Table 3).

#### 3.4.3. Beta Diversity Index

Jaccard similarity matrices were visualized using non-metric multidimensional scaling (NMDS). At the genus level, samples were clustered according to Treatment (NMDS stress = 0.13419). Control samples were grouped separately from the experimental groups, II, III, and IV. The ANOSIM analysis showed significant differences, although R values were low (R = 0.18067; *p* < 0.008) (Figure 9).

#### 3.4.4. Bacterial Functional Profile Based on KEGG Pathway Analysis

In the shrimp gut samples, six functional categories were found; that is, metabolism (60.03–61.05%), genetic information processing (10.88–11.01%), human diseases (11.43–11.94%), environmental information processing (1.39–1.59%), cellular processes (6.90–738%), and organismal systems (8.10–8.27%). No significant differences were found in the Treatments (bacteria in the water) compared to the control (*p* < 0.05) (Table 4).

### 3.5. Correlation between Functional Categories and Immune and Productive Variables

Only the metabolism of linoleic acid and quorum sensing significantly correlate with productive variables in the KEGG pathway analysis at Level 3. Figure 10 shows the positive correlation between the metabolism of linoleic acid and superoxide anion (*p* = 0.045), as well as the final weight (*p* < 0.05). Figure 11 shows the positive correlation between quorum sensing and shrimp survival (*p* < 0.05).

## 4. Discussion

The high animal densities that are managed in aquaculture systems deteriorate the water quality, which generates stress and susceptibility to diseases [42,43]. Ammonia and nitrites at high concentrations are toxic to aquaculture animals, and their toxicity depends on temperature, salinity, pH, or the developmental stage of the organism [44,45]. Furthermore, aquatic nitrogenous wastes, such as nitrites, can directly affect the nitrite content in shrimp intestine [46]. Therefore, reducing nitrogenous waste in culture systems is necessary for producing cultured animals and preventing deterioration of the surrounding environment [47].

The bacterial consortia, composed of several probiotic strains, usually have synergy, leading to a bioremediation effect to improve water quality [48]. Some bacterial strains contain enzymes that oxidize ammonia and nitrites, contributing to the nitrification process of nitrogenous waste in culture systems [49]. In this work, only bacilli oxidizes TAN and nitrites while reducing nitrates in vitro since nitrogen bubbles were produced, indicating a nitrification and denitrification process. The addition of bacilli and LAB to the water decreased TAN and nitrite concentrations, showing a nitrifying effect, so a denitrifying effect was not observed as occurred in vitro with bacilli. Also, it is important to mention that in the control, there was also a nitrification process as TAN and nitrites were not high (0.3 and 0.4 mg L^−1^) and nitrates were 2.47 mg L^−1^, which could be caused by the natural bacteria community from the shrimp and initial filtered water. The nitrification process was reported in the culture of *L. vannamei* [50,51] when *Bacillus subtilis* L10 and G1 were inoculated into the water. The denitrification process was observed in *Bacillus* sp. SC16 in an intensive fishery aquaculture pond [52]. Finally, there are no reports indicating *Pediococcus* and *Leuconostoc* as nitrifying bacteria.

In the aquatic environment, bacterial components in the sediment and water affect the bacterial communities in the shrimp intestine [53,54]. Zhou et al. [55] mention that 35.32% of bacteria in the intestine of the white shrimp come from water and 54.58% from sediments. The intestinal microbiota of organisms affects their digestion, absorption, growth, and immune response [56]. Regarding the effect of bacteria and nitrogenous waste on growth, the best shrimp growth was observed in the Treatments with bacilli or the mixture of bacilli and LAB in the water. It is known that bacterial species, such as *Bacillus*, inoculated into the water, improve the growth of *L. vannamei* [50,57] thanks to the increase in the production of enzymes that digest feed nutrients [58] and short-chain fatty acids (SCFAs) [59]. On the other hand, elevated nitrites affected the growth of *L. vannamei* [60,61] and reduced food consumption in *Farfantepenaeus brasiliensis* [62]. Similarly, Han et al. [63] mention that ammonia and nitrites reduce the growth in weight and length of *L. vannamei* cultured in the laboratory.

The optimal concentration of nitrogen in water in the form of ammonium (NH_4_^+^), ammonia (NH_3_), nitrites (NO_2_^−^), and nitrates (NO_3_^−^) is 0.2−2.0, 0.09−0.11, <0.23, and 0.2−10 mg·L^−1^, respectively [64,65]. Boardman et al. [66] mention that crustaceans excrete ammonia and cannot convert it to less toxic compounds, so the high concentration of these compounds is toxic to crustaceans through gill absorption. In the present work, TAN’s concentration was below the toxic concentration. However, the stress caused by ammonia reduces the survival of *L. vannamei* grown in the laboratory [63,67]. Nitrites are toxic to crustaceans because they convert hemocyanin into meta-hemocyanin, which is incapable of transporting oxygen [68]. In our work, nitrites were in the range of 0.40 and 0.46 mg·L^−1^ above the optimal concentration in all Treatments. Therefore, it is possible that mortality was caused, in part, due to the stress induced by nitrites. The nitrification driven by bacteria (bacilli and LAB) could thus protect *L. vannamei* against the negative effect of nitrites. In the work of Furtado et al. [60], a high mortality of *L. vannamei* was observed due to nitrite poisoning (5.0−40.0 mg L^−1^), and they mention that, during molting, there is a high consumption of oxygen, so nitrites cause hypoxia or metabolic anoxia. In contrast, Huang et al. [61] did not observe mortalities in *L. vannamei* exposed to high concentrations of nitrites (2.0, 6.67, and 20 mg L^−1^).

Invertebrates possess a natural, non-specific immune response, which plays an important role in resistance to microbial diseases [66] as they do not produce antibodies like vertebrates [69]. In our work, no significant change was observed in the number of hemocytes in shrimp treated with bacteria in the water and the accumulation of nitrogenous waste. Conversely, Tseng and Chen [70] found that nitrite accumulation decreases the number of hemocytes in *L. vannamei*.

Regarding the superoxide anion, it increased in the hemocytes of the shrimp where nitrites were above the optimal range (0.09−0.11 mg L^−1^). Liao et al. [71] found that the concentration of nitrites in the hemolymph of *L. vannamei* was similar to that of the culture water and that this exposure increased the generation of ROS during the respiratory burst (measured indirectly by the high activity of catalase in the hepatopancreas). In contrast, Huang et al. [61] found no changes in SOD activity in the hepatopancreas of *L. vannamei* exposed to high nitrite concentrations, and Cheng et al. [67] found similar results in *L. vannamei* exposed to ammonia during 24 h. Regarding phenoloxidase, its activity decreased in the Treatments with LAB alone and the mixture of bacteria (bacilli and LAB) in the water. The decrease in phenoloxidase activity could indicate an optimal physiological state with less biotic stress.

In the shrimp intestine, the microbiota is complex and variable, mainly affected by diet, developmental phase, immune response, metabolism, and the environment surrounding the animals [72,73]. At the phylum level, proteobacteria and bacteroidota predominated. However, no significant differences were observed in bacterial Treatments. In the work of Huang et al. [61], at the concentration of 2 mg L^−1^ of nitrites, the highest relative abundance corresponded to bacteroidota, followed by proteobacteria and actinobacteria. At the highest nitrite concentration (20 mg L^−1^), the highest abundance corresponded to proteobacteria, followed by bacteroidota and actinobacteria. All the above demonstrated that proteobacteria and bacteroidota are the most important components in shrimp intestines under this condition. Proteobacteria is a core member of shrimp gut microbiota [74]. According to Xiong et al. [75] and Rungrassamee [76], the abundance of this phylum indicates efficient colonization of the shrimp intestinal epithelium and possible degradation of agar and cellulose, as well as nitrogen fixation in the shrimp rectum [77]. Regarding the phylum bacteroidota, it increases when the amount of dietary protein and fat increases [78,79] and has a very important role due to its ability to utilize nitrogenous waste, biotransform steroids, and ferment carbohydrates [67,72,79,80].

Within the native microbiota that predominates in marine species, the genus *Vibrio* constitutes the greatest abundance [81]. In our study, the highest relative abundance was presented by the *Vibrio* genus, with the highest abundance in the bacilli Treatment, followed by the Treatment with LAB in the water. Intriago et al. [17] and Vega–Carranza [82] reported the *Vibrio* genus as abundant in white shrimp intestines in Treatments supplementing *Bacillus* species to the culture water, which coincides with the report of Zheng et al. [83], where the mentioned genus predominated in most Treatments supplemented with different *Bacillus* mixtures. The control of *Vibrio* in shrimp culture is very important as it could affect shrimp health [84]. However, some *Vibrio* strains are beneficial to shrimp health such as *V. hepatarius* and *V. diabolicus*, which protect *Penaeus vannamei* larvae against *V. parahaemolyticus* [85] or *V. parahaemolyticus*, *V. diazotrophicus*, *V. natriegens*, and *V. campbellii* strains that utilize several organic carbon sources (unused organic matter) and can fix nitrogen [86,87]. The second-most abundant genera was *Pseudoalteromonas*, which is a probiotic bacteria that is antagonistic against *V. parahaemolyticus*, *V. harveyi*, and *V. nigripulchritudo* [88,89].Bioencapsulated *Pseualteromonas* in *Artemia* sp. increases the immune response of shrimp and their resistance to infections caused by *V. harveyi* [90,91].

Functional redundancy provided by high microbial diversity allows an ecosystem to be more stable and resistant to stress [92,93]. The total bacterial species richness in a sample can be determined with the Chao1 and ACE alpha indices [94,95]. On the other hand, Shannon and Simpson alpha indices consider the richness of the microbial community and the evenness (relative abundance of different species) [96,97]. The Treatment with bacilli in the water showed lower species richness (alpha diversity) compared to the control and the other Treatments. Zhou et al. [55] found that the intestine of *L. vannamei* cultured in a nutrient-rich environment (shrimp feces and organic waste) showed higher bacterial diversity. It is possible that the Treatment with bacilli in the water had a lower nutrient load and, although it presented a lower richness of bacteria, it did show greater survival. Beta diversity is the degree of change or replacement in species composition between different communities [98]. Regarding the analysis of diversity among communities (NMDS), this showed a grouping in each Treatment. However, the bacterial community of the control was significantly different from the community of the Treatments with bacteria in the water, where it was similar. In contrast, Landsman et al. [99] found that the bacterial community in the intestines of shrimp cultured in an indoor culture system showed no variations. Similarly, Vega–Carranza et al. [100] found that the intestine of *L. vannamei* fed with synbiotics and postbiotics of bacilli and vibrio showed similar bacterial communities.

The potential function of shrimp gut bacterial microbiota can be predicted using the KEGG database [101]. The functional profile has a fundamental role in the ecological balance of intestinal microbiota. However, it is important to mention that the predominant function is metabolism [102,103]. Wang et al. [104] mentioned that overrepresented bacterial metabolism may be related to energy consumption to satisfy the physiological activities of the host (shrimp). In this work, the bacterial functional profile did not change significantly by the Treatments. However, linoleic acid metabolism (fatty acid) was positively correlated with final weight. Fatty acids play a fundamental role in cell structure and cell homeostasis [105] and are a source of energy stored in triacylglycerols [106]. It is known that fatty acids modulate the immune response thanks to their influence on the structure, function, metabolism, surface proteins, and intracellular receptors of cells [105]. Linoleic acid stimulates the production of ROS by activating the NADPH oxidase enzyme in rat fibroblasts [107]; it agrees with the positive correlation of the immune response as revealed by the detection of superoxide anion in the present study.

Quorum sensing (QS) is a cell-to-cell communication system that regulates biofilm formation and the expression of virulence genes [108,109]. In this work, QS was not significantly different among Treatments. However, a positive correlation with shrimp survival was observed; it suggests a benefit to its health. Similarly, QS had a health benefit on farmed *L. vannamei* shrimp [109] and the farmed turbot (*Scophthalmus maximus*) [108].

## 5. Conclusions

In hyperintensive culture systems, high stocking density produces excess nitrogenous wastes, which deteriorate water quality, affecting the physiology and gut microbiota of shrimp. The results of this study showed that bacilli and LAB in the water of hyper-intensive culture systems act as heterotrophic nitrifiers, modulate the intestinal microbiota and immune response, and improve the growth and survival of shrimp. To our knowledge, this is the first report on *Pediococcus pentosaceus* and *Leuconostoc mesenteroides* as nitrifying bacteria.

## Figures and Tables

**Figure 1 animals-14-02676-f001:**
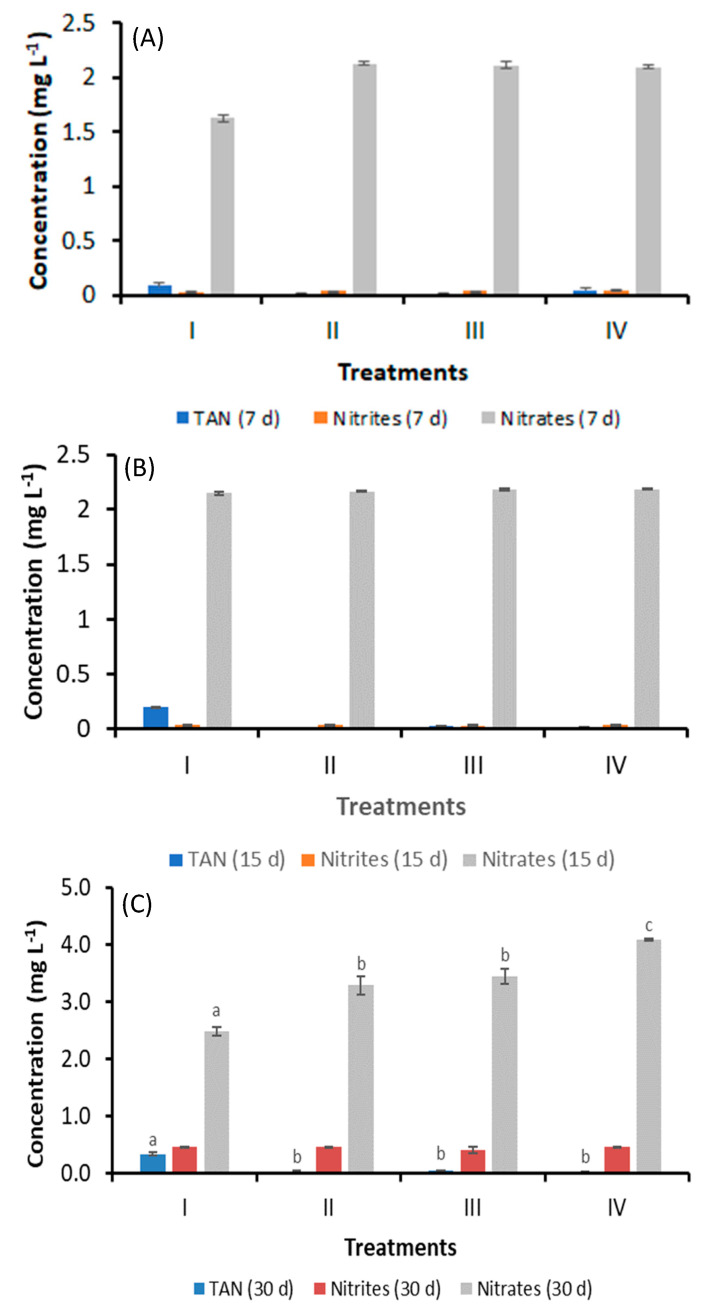
Concentration of TAN, nitrites, and nitrates on Days 7 (**A**), 15 (**B**), and 30 (**C**) in the shrimp culture system without water exchange and treated with bacteria. Treatments: (I) Control without bacteria in the water; (II) bacilli in the water; (III) LAB in the water; (IV) bacilli + LAB in the water. Data are mean ± SD. Different letters indicate significant differences.

**Figure 2 animals-14-02676-f002:**
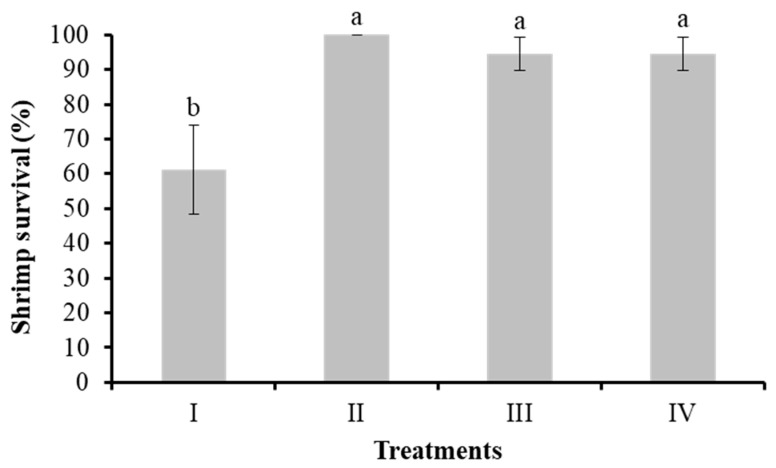
Survival of shrimp cultured without water exchange and treated with bacteria. Treatments: (I) Control without bacteria in the water; (II) bacilli in the water; (III) LAB in the water; (IV) bacilli + LAB in the water. Data are mean ± SD. Different letters indicate significant differences.

**Figure 3 animals-14-02676-f003:**
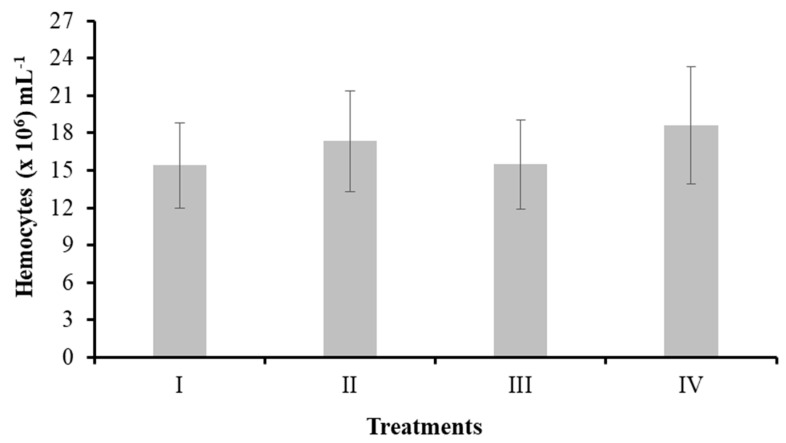
Total hemocyte count in *L. vannamei* cultured without water exchange and treated with bacteria. Treatments: (I) Control without bacteria in the water; (II) bacilli in the water; (III) LAB in the water; (IV) bacilli + LAB in the water. Data are mean ± SD.

**Figure 4 animals-14-02676-f004:**
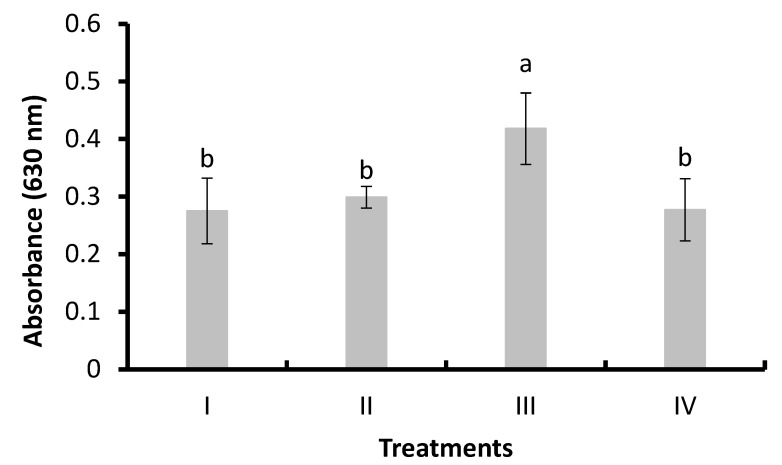
Superoxide anion in hemocytes of *L. vannamei* cultured without water exchange and treated with bacteria. Treatments: (I) Control without bacteria in the water; (II) bacilli in the water; (III) LAB in the water; (IV) bacilli + LAB in the water. Data are mean ± SD. Different letters indicate significant differences.

**Figure 5 animals-14-02676-f005:**
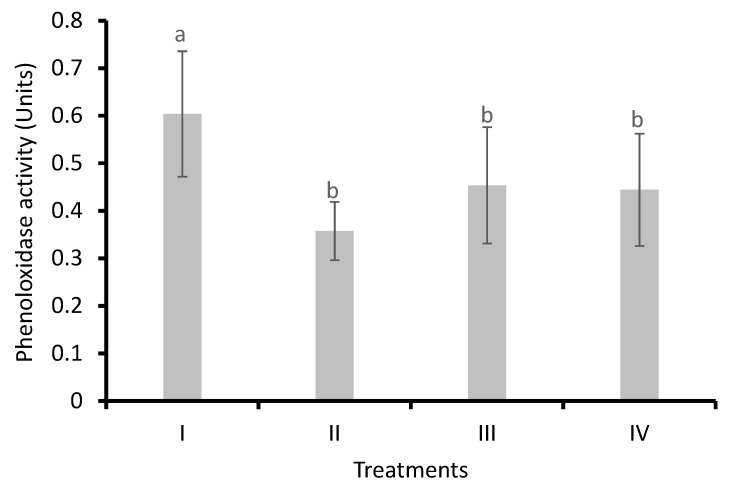
Phenoloxidase activity (absorbance) in hemolymph of *L. vannamei* cultured without water exchange and treated with bacteria. Treatments: (I) Control without bacteria in the water; (II) bacilli in the water; (III) LAB in the water; (IV) bacilli + LAB in the water. Data are mean ± SD. Different letters indicate significant differences.

**Figure 6 animals-14-02676-f006:**
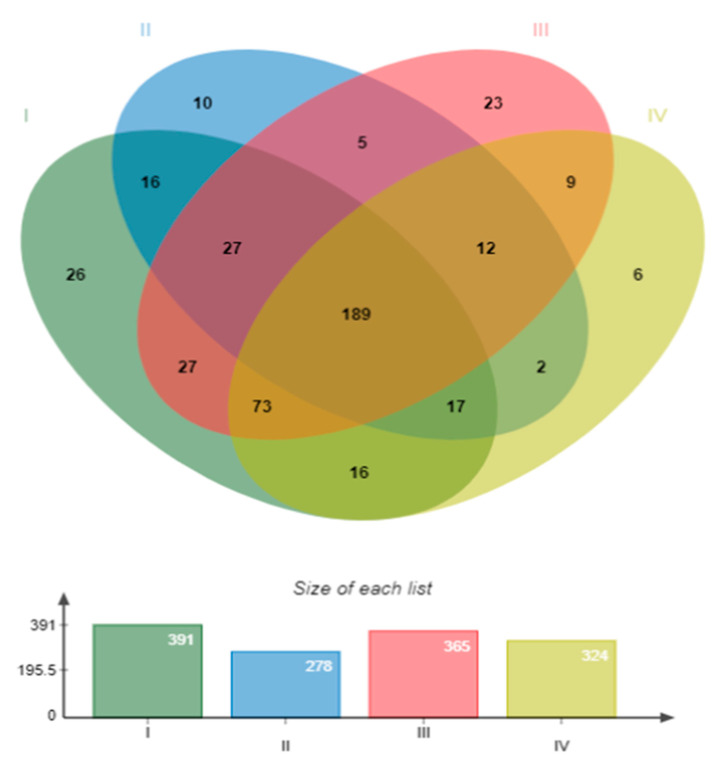
Venn analysis of the bacterial communities in the shrimp intestine at the OTUs level. Treatments: (I) Control without bacteria in the water; (II) bacilli in the water; (III) LAB in the water; (IV) bacilli + LAB in the water.

**Figure 7 animals-14-02676-f007:**
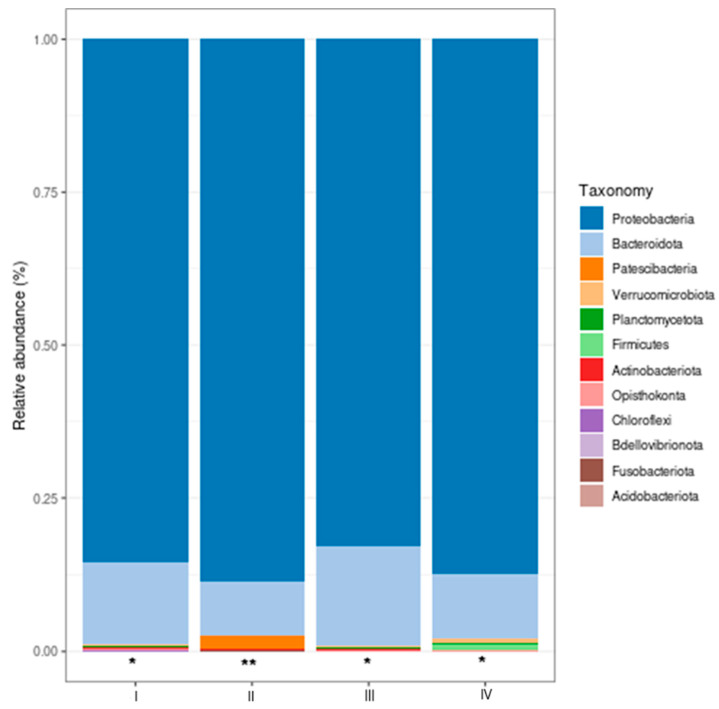
Most abundant bacterial phyla (%) in the shrimp intestine. Treatments: (I) Control without bacteria in the water; (II) bacilli in the water; (III) LAB in the water; (IV) bacilli + LAB in the water. Patescibacteria phylum (* no significant differences [*p* > 0.05], ** significant differences [*p* < 0.05]). The analysis was done with Shaman.

**Figure 8 animals-14-02676-f008:**
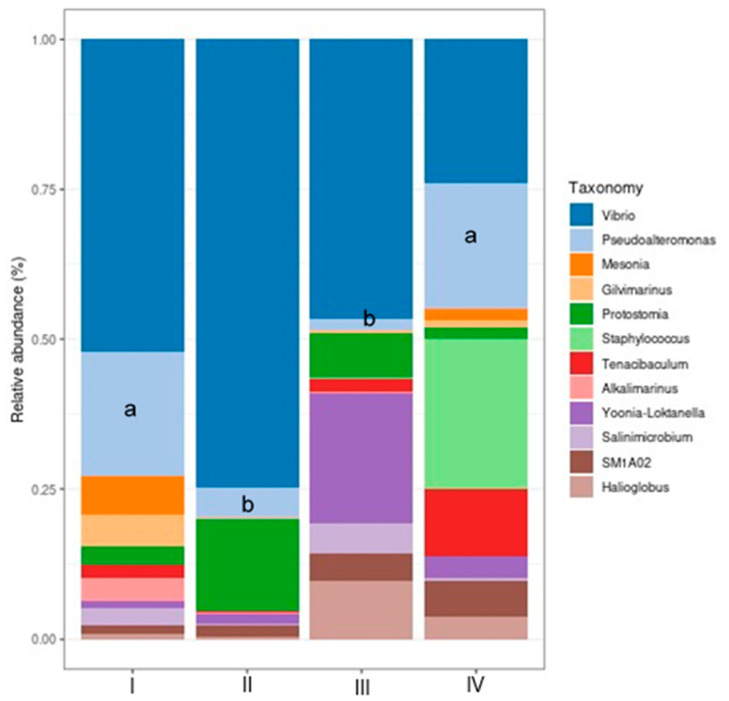
Most abundant bacterial genera (%) in the shrimp intestines. Treatments: (I) Control without bacteria in the water; (II) bacilli in the water; (III) LAB in the water; (IV) bacilli + LAB in the water. Different letters indicate significant differences (*p* < 0.05). The analysis was done with Shaman.

**Figure 9 animals-14-02676-f009:**
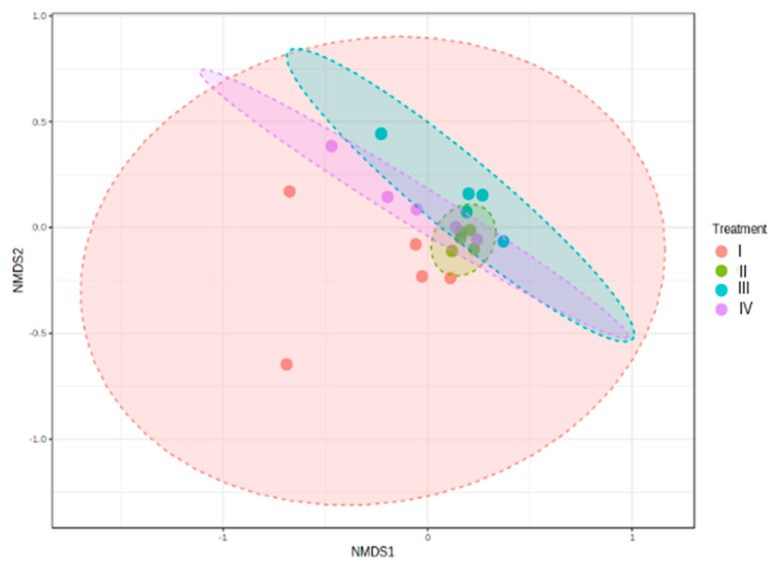
Beta diversity of intestinal microbiota of *L. vannamei* at the genus level using non-metric multidimensional scaling based on Jaccard distances in MicrobiomeAnalyst. Treatments: (I) Control without bacteria in the water; (II) bacilli in the water; (III) LAB in the water; (IV) bacilli + LAB in the water. ANOSIM test, *p* < 0.008.

**Figure 10 animals-14-02676-f010:**
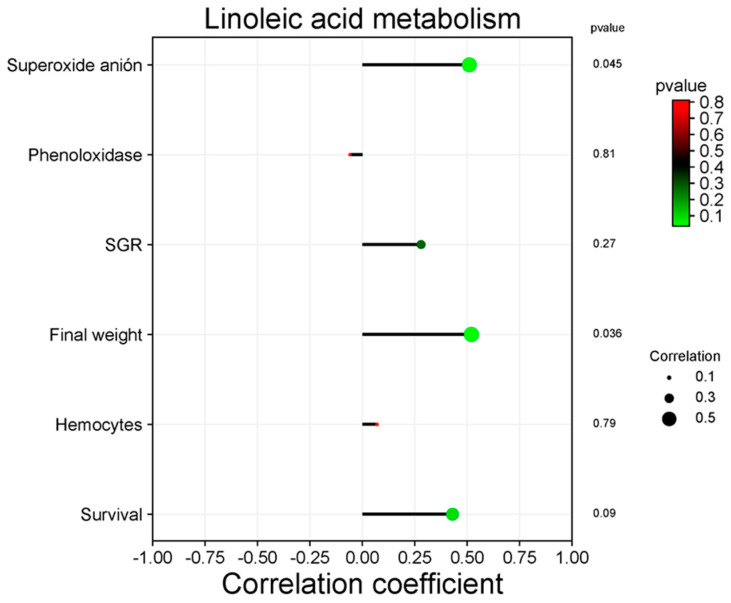
Correlation among linoleic acid metabolism of intestinal bacteria of *L. vannamei* and immune and productive variables. Spearman correlation analysis.

**Figure 11 animals-14-02676-f011:**
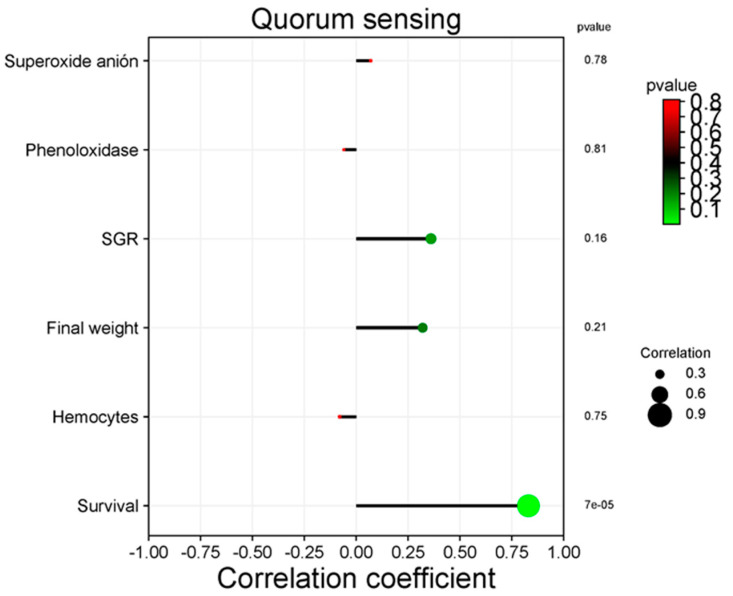
Correlation among quorum sensing of intestinal bacteria of *L. vannamei* and immune and productive variables. Spearman correlation analysis.

**Table 1 animals-14-02676-t001:** Physicochemical parameters in shrimp culture system without water exchange and treated with bacteria. Treatments: (I) Control without bacteria in the water; (II) Bacilli in the water); (III) LAB in the water; and (IV) Bacilli + LAB in the water. Values = mean ± SD.

Treatment	DO (mg mL^−1^)	pH	T (°C)	S (PSU)
Control (I)	5.2 ± 0.05	8.2 ± 0.05	29.8 ± 0.3	30 ± 0.03
II	5.2 ± 0.01	8.2 ± 0.02	30.0 ± 0.4	30 ± 0.02
III	5.2 ± 0.04	8.2 ± 0.04	29.8 ± 0.5	30 ± 0.04
IV	5.3 ± 0.08	8.2 ± 0.06	30.0 ± 0.3	30 ± 0.03
Optimal range	4 to 10	8.1 to 9	23 to 30	15 to 35

**Table 2 animals-14-02676-t002:** Growth of shrimp cultured without water exchange and treated with bacteria. Treatments: (I) Control without bacteria in water; (II) bacilli in the water); (III) LAB in the water; (IV) bacilli + LAB in the water. Data are mean ± SD. Different letters indicate significant differences.

Shrimp Growth	I	II	III	IV
Initial weigth (g)	0.7 ± 0.05	0.79 ± 0.06	0.73 ± 0.05	0.77 ± 0.07
Final weigth (g)	3.09 ± 0.27 ^b^	4.18 ± 0.53 ^a^	3.87 ± 0.23 ^ab^	4.10 ± 0.09 ^a^
SGR (%d^−1^)	4.16 ± 0.32	4.75 ± 0.25	4.76 ± 0.33	4.78 ± 0.35

**Table 3 animals-14-02676-t003:** Alpha diversity indices in the treatment groups and the control derived from MicrobiomeAnalyst. Treatments: (I) Control without bacteria in the water; (II) bacilli in the water; (III) LAB in the water; (IV) bacilli + LAB in the water. The mean ± SD are indicated. Different letters indicate significant differences.

Indices	I	II	III	IV
Shannon	1.71 ± 0.62	1.22 ± 0.18	1.19 ± 0.53	1.47 ± 0.35
Simpson	0.60 ± 0.16	0.53 ± 0.13	0.40 ± 0.21	0.47 ± 0.13
Chao1	149.74 ± 27.71 ^a^	93.95 ± 25.25 ^b^	137.03 ± 10.27 ^a^	137.10 ± 14.54 ^a^
ACE	146.34 ± 27.87 ^a^	88.95 ± 28.97 ^b^	132.22 ± 15.70 ^ab^	115.50 ± 45.48 ^ab^

**Table 4 animals-14-02676-t004:** The KEGG functional categories at Level 1 (iVikodak) of microbiota found in the shrimp gut.

Treatment	Metabolism (%)	GIP (%)	HD (%)	EIP (%)	CP (%)	OS (%)
Control	61.05 ± 1.22	11.01 ± 0.15	11.43 ± 0.63	1.39 ± 0.23	6.90 ± 0.58	8.19 ± 0.11
Bacilli in water	60.03 ± 0.14	10.92 ± 0.05	11.94 ± 0.04	1.59 ± 0.01	7.38 ± 0.04	8.10 ± 0.01
LAB in water	60.19 ± 0.44	10.88 ± 0.13	11.87 ± 0.17	1.55 ± 0.10	7.22 ± 0.35	8.27 ± 0.32
Bacilli and LAB in water	60.16 ± 0.39	11.00 ± 0.05	11.85 ± 0.17	1.49 ± 0.14	7.21 ± 0.22	8.27 ± 0.22

GIP = genetic information processing; HD = human diseases; EIP = environmental information processing; CP = cellular processes; OS = organismal systems.

## Data Availability

The sequences were submitted to the NCBI (PRJNA1044443). The rest of the data are available upon request.

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
