# Peer review of "Investigating the Effect of Bacilli and Lactic Acid Bacteria on Water Quality, Growth, Survival, Immune Response, and Intestinal Microbiota of Cultured Litopenaeus vannamei"

_animals, 2024, doi:10.3390/ani14182676_

Round 1
Reviewer 1 Report
Comments and Suggestions for Authors
Reviewers' Comments to Authors:
The manuscript entitled “Investigating the effect of bacilli and lactic acid bacteria on water quality, growth, survival, immune response and intestinal microbiota of cultured Litopenaeus vannamei” by Vega-Carranza et al. demonstrates the scientific attempts to explain the efficacy of probiotic application on growth, immune response and gut microbiota of Pacific white shrimp.
Based on scientific consideration, the manuscript contains some interesting findings that contribute to knowledge of shrimp disease and immunology and can be further applicable to the shrimp aquaculture industry. However, it does contain many errors and unclear points that the authors must pay more attention to address and improve the quality of this manuscript to meet an acceptable high-standard journal.
The critical points of major and minor issues that the authors must pay more attention to to improve the quality of this vital manuscript are as follows.
Simple Summary
This part is too short and not describe much about the essence of the current research works.
Abstract
Line 27.
1) Italicize “In vitro” and all Italic contents throughout the manuscript.
2) Full form of Bacillus genus of “B. licheniformis”.
Line 28. Specific bacterial species of “bacilli” and “LAB” should be firstly indicated.
1. Introduction
Line 51. Correct a grammatical error of “…is generated, which…”.
Line 77. Correct the reference “(Intriago et al. 2018)” to numeric system and please correct the serial of reference number throughout.
At the end of this section, the out put/out come of the current research should be described.
2. Materials and Methods
2.1. Bacillus licheniformis BCR 4-3 and lactic acid bacteria (LAB) culture
Line 92. Correct “5000 g” to “5000 ×g” and every place throughout.
2.2. In vitro utilization of ammonium, nitrites and nitrates by bacilli and LAB
Line 102. Correct “(NH4)2SO4, “KNO3”, “MgSO4”, “KH2PO4”.
Line 104. Correct “(Smooth culture tube)”.
2.3.1. Origin and acclimatization of experimental shrimp
Line 113. Declare salinity levels used to acclimatize the experimental shrimp.
Line 114. I was not clear about “chlorine treatment”!! Please properly clarify.
Line 116. Please be consistent in using either “PSU or ppt” throughout the manuscript.
2.3.2. Bioassay: Effect of bacilli and LAB on water quality, growth, survival, immune system, and intestinal microbiota of shrimp
Much information of this part is very important, but the description in some part is very unclear and poor description.
1) The authors should declare the exact replication used in each treatment.
2) The authors always mention the problems caused by high stocking density. Technically, the authors applied 12 shrimp/tank/20 L. Therefore, the authors should describe, did this setup corresponded to the high stocking density that the authors targeted to test.
3) It was not unclear in treatments III and VI in which two LAB of “Pediococcus pentosaceus, and Leuconostoc mesenteroides” were applied. The authors should clearly declare how much concentration of each bacterium was used in each treatment.
4) Line 139. The term “nitrifying bacteria” should be avoided.
5) Verify “(2:2:1)”.
6) Line 144. The following content should be properly verified; “The samples (five per treatment) were sent to the Research Center for 144 Food and Development (CIAD, Mazatlán, Sinaloa, Mexico).”, since it is too brief to understand. Please clarify what the authors do and what was the objectives of this part.
7 Growth parameters should be indicated with more parameters. FCR and ADG should be demonstrated.
2.3.3. Hemolymph collection
Line 154. Correct “[27}”.
2.3.4. Hemocyte count
Line 159. Declare the model and company of the compound microscope used.
2.3.5. Superoxide anion
Line 169-171. Please revise the following wrong description of the specific protocol; “The optical density of the dissolved formazan was read at 630 nm in a Thermo Spectronic Genesys 2 Spectrophotometer 2.10 (Thermo Fisher Scientific).”. After DMSO and KOH were added, it must be centrifuged and supernatant must be removed to the new tube to prevent disturbance of the remaining debris, before determining the absorbance in the obtained solution.
And please correct “The absorbance of the dissolved formazan was read at the optical density (OD) at 630 nm”.
2.3.6. Phenoloxidase activity (PO) in hemolymph
Line 182-183. This information is wrong; “Enzyme activity was expressed as absorbance at 492 nm”. After absorbance reading, the authors must calculate the “unit enzyme/mL” to express the concentrations of detected enzymes.
2.4. Water sampling, TAN, nitrites, and nitrates determination
In this part, the authors described that 1000 mL was collected for water analysis. However, in section “2.3.2”, the authors described that water parameters were measured every week (7 day). Considerably, the authors used 20 L/tank to raise experimental shrimps, therefore it must be at least 7 L must be loosed. Is this information true? And if yes, it must be seriously affected to the environment of experimental conditions.
Additionally, water quality parameters in each week should be illustrated.
2.5.2. Gut bacterial taxonomy, abundance, and diversity analysis
Line 216. Italicize “16S rRNA”.
Line 229. Correct “FO activity”.
Additional comments.
To increase the flow of this part, a challenge test should be conducted with some pathogens. This will clear the hypothesis of the efficacy of all application probiotics.
3. Results
3.1. In vitro utilization of ammonium, nitrites and nitrates by bacilli and LAB
In this part, the authors should give more description, of how the authors known bobbles in the Duram tube are nitrogen gas??
3.2.2. Shrimp survival
Survival rate in each week and the reasons of shrimp death should be declared.
3.3. Immune system. Please change this topic to “3.3. Immune parameters”.
3.3.1. Total hemocyte count
Line 288. Please be consistent in using a proper unit of “per milliliter”.
Figure 4. Technically, superoxide anion is measured as “absorbance” to indicate formazan concentration and indicated as an absorbance unit. Therefore, it is not possible that the absorbance will be too high as shown in this figure.
Line 299. Change “hemolymph” to “hemocytes”.
3.3.3. Phenoloxidase activity in shrimp hemolymph
Figure 5. The unit of PO activity must be indicated as “unit enzyme/mL”. Please correct the graph and description in the text.
Line 312. Italicize “L. vannamei”.
3.4.1. Relative bacterial abundance in the shrimp gut
Line 331. Correct “bacteri”.
Line 341. Italicize “Mesonia”.
3.4.3. Beta diversity index
Line 377. Italicize “L. vannamei”.
4. Discussion
There are many errors and unnecessary references were cited. Most information of this section must be properly overhauled.
Line 415. The logic of the following content is wrong; “…as this organ is exposed to the water environment [45].”.
Line 433. Correct “[54] mention” to “Zhou et al. [54] mention” and the other places in the manuscript.
Line 465. The following content is unnecessary; “[69] found that L. vannamei treated with 400 and 600 mg·L-1 of Gracilaria tenuistipitata extract for 3 h and then exposed to 5 mg·L-1 of ammonia, increased hemocytes compared to 466 the control group.”. The authors should step forward to describing the benefits of bacteria that properly reduce ammonia in culture water.
Similarly, the following contents must be removed.
1) Line 472-428. Similaly, [69] found that in the hemolymph of shrimp treated with G. tenuistipitata extract and then exposed to ammonia, the activity of the enzyme superoxide dismutase increased, which has the superoxide anion radical as a substrate. In contrast, [61] found no changes in SOD activity in the hepatopancreas of L. vannamei exposed to high nitrite concentrations, and [71] found similar results in L. vannamei pretreated with dietary inositol and exposed to ammonia during 24 h. Regarding phenoloxidase, its activity decreased in the treatments with LAB alone and the mixture of bacteria in the water. The decrease in phenoloxidase activity could indicate an optimal physiological state with less biotic stress. Converseley, data reported by [69] showed an increase in the activity of the enzyme in shrimp treated with G. tenuistipitata and ammonia.
2) Line 486-490. In the work of [61], at the concentration of 2 mg L-1 of nitrites, the highest relative abundance corresponded to Bacteroidota, followed by Proteobacteria and Actinobacteria. At the highest nitrite concentration (20 mg L-1), the highest abundance corresponded to Proteobacteria followed by Bacteroidota and Actinobacteria.
Line 545 and 550. Correct grammatical errors of “which”.
5. Conclusion
As suggested in an earlier section, the term “nitrifying bacteria” should be avoided.
References
1) Please carefully correct inconsistent formats and other errors in some references. There are many errors in this part.
2) There are too many references used in this part. The authors should carefully reduce all necessary references in this part.
Comments on the Quality of English Language
-
Author Response
Reviewer 1
Simple Summary
-This part is too short and not describe much about the essence of the current research works.
We change the simple summary.
Abstract
Line 27.
1) Italicize “In vitro” and all Italic contents throughout the manuscript.
It was done.
2) Full form of Bacillus genus of “B. licheniformis”.
It was done.
Line 28. Specific bacterial species of “bacilli” and “LAB” should be firstly indicated.
It was done.
- Introduction
Line 51. Correct a grammatical error of “…is generated, which…”.
It was done.
Line 77. Correct the reference “(Intriago et al. 2018)” to numeric system and please correct the serial of reference number throughout.
It was done. In addition, the numbering of the citations was changed because some were deleted.
At the end of this section, the out put/out come of the current research should be described.
We include the following information: This study will allow us to better understand the effect of heterotrophic bacteria such as bacilli and LAB on water quality and shrimp physiology and gut microbiota.
- Materials and Methods
2.1. Bacillus licheniformis BCR 4-3 and lactic acid bacteria (LAB) culture
Line 92. Correct “5000 g” to “5000 ×g” and every place throughout.
It was done.
2.2. In vitro utilization of ammonium, nitrites and nitrates by bacilli and LAB
Line 102. Correct “(NH4)2SO4, “KNO3”, “MgSO4”, “KH2PO4”.
It was done.
Line 104. Correct “(Smooth culture tube)”.
We put Durham tube.
2.3.1. Origin and acclimatization of experimental shrimp
Line 113. Declare salinity levels used to acclimatize the experimental shrimp
Line 119. 30 PSU salinity.
Line 114. I was not clear about “chlorine treatment”!! Please properly clarify.
Water was treated with liquid chlorine (1.5 mL L-1).
Line 116. Please be consistent in using either “PSU or ppt” throughout the manuscript.
Ok.
2.3.2. Bioassay: Effect of bacilli and LAB on water quality, growth, survival, immune system, and intestinal microbiota of shrimp
Much information of this part is very important, but the description in some part is very unclear and poor description.
1) The authors should declare the exact replication used in each treatment.
Each treatment with 3 replicates (line 129).
2) The authors always mention the problems caused by high stocking density. Technically, the authors applied 12 shrimp/tank/20 L. Therefore, the authors should describe, did this setup corresponded to the high stocking density that the authors targeted to test.
This setup corresponded to 600 shrimp m3 (line 126).
3) It was not unclear in treatments III and VI in which two LAB of “Pediococcus pentosaceus, and Leuconostoc mesenteroides” were applied. The authors should clearly declare how much concentration of each bacterium was used in each treatment.
Each bacteria (3 × 106 CFU·L-1)..line 131.
4) Line 139. The term “nitrifying bacteria” should be avoided.
In other Works, it has been shown that Bacillus licheniformis has the enzymes to be able to utilize TAN, nitrites and nitrates. According to the results, we can say that the bacilli and LAB acted as nitrifiers.
5) Verify “(2:2:1)”.
It is okay. We had 3 tanks and took 2, 2, and 1 shrimp from de last tank.
6) Line 144. The following content should be properly verified; “The samples (five per treatment) were sent to the Research Center for 144 Food and Development (CIAD, Mazatlán, Sinaloa, Mexico).”, since it is too brief to understand. Please clarify what the authors do and what was the objectives of this part.
The samples (five per treatment) were sent to the Research Center for Food and Development (CIAD, Mazatlán, Sinaloa, Mexico) for bacterial DNA extraction, library preparation, and sequencing in Illumina MiniSeq.
7 Growth parameters should be indicated with more parameters. FCR and ADG should be demonstrated.
We were unable to obtain the FCA because we did not do water changes and we could not obtain the leftover feed (it was very little feed) because we wanted there to be nitrogenous waste.
2.3.3. Hemolymph collection
Line 154. Correct “[27}”.
It was done.
2.3.4. Hemocyte count
Line 159. Declare the model and company of the compound microscope used.
Labomed, Labo America, Inc., Fremont, CA, USA (line 166).
2.3.5. Superoxide anion
Line 169-171. Please revise the following wrong description of the specific protocol; “The optical density of the dissolved formazan was read at 630 nm in a Thermo Spectronic Genesys 2 Spectrophotometer 2.10 (Thermo Fisher Scientific).”. After DMSO and KOH were added, it must be centrifuged and supernatant must be removed to the new tube to prevent disturbance of the remaining debris, before determining the absorbance in the obtained solution.
And please correct “The absorbance of the dissolved formazan was read at the optical density (OD) at 630 nm”.
The samples were centrifuged and the supernatant was placed in new tubes. The optical density of the dissolved formazan was measured at 630 nm (OD630) (lines 177-180).
2.3.6. Phenoloxidase activity (PO) in hemolymph
Line 182-183. This information is wrong; “Enzyme activity was expressed as absorbance at 492 nm”. After absorbance reading, the authors must calculate the “unit enzyme/mL” to express the concentrations of detected enzymes.
We only used the absorbance of the colorimetric reaction to determine the enzyme activity. We did not use the protein to derive the enzyme activity units because in shrimp, hemocyanin contributes more than 85% of the total protein.
2.4. Water sampling, TAN, nitrites, and nitrates determination
In this part, the authors described that 1000 mL was collected for water analysis. However, in section “2.3.2”, the authors described that water parameters were measured every week (7 day). Considerably, the authors used 20 L/tank to raise experimental shrimps, therefore it must be at least 7 L must be loosed. Is this information true? And if yes, it must be seriously affected to the environment of experimental conditions.
The water that was taken was replaced with clean water (line 196).
Additionally, water quality parameters in each week should be illustrated.
Only three water samples were taken on day 7, 15 and 30 (see information on line 134)...the results of each sample were included.
2.5.2. Gut bacterial taxonomy, abundance, and diversity analysis
Line 216. Italicize “16S rRNA”.
It was done.
Line 229. Correct “FO activity”.
It was done.
Additional comments.
To increase the flow of this part, a challenge test should be conducted with some pathogens. This will clear the hypothesis of the efficacy of all application probiotics.
We consider that the challenge with pathogens is not part of the objective of this work.
- Results
3.1. In vitro utilization of ammonium, nitrites and nitrates by bacilli and LAB
In this part, the authors should give more description, of how the authors known bobbles in the Duram tube are nitrogen gas??
Nitrogen bubbles were produced from the chemical substances provided in the culture médium (line 253).
3.2.2. Shrimp survival
Survival rate in each week.
We consider it unnecessary to record survival every week because shrimp mortality started on day 34 and the bioassay ended on day 35.
The reasons of shrimp death should be declared.
We addressed that point in the discussion.
3.3. Immune system. Please change this topic to “3.3. Immune parameters”.
It was done.
3.3.1. Total hemocyte count
Line 288. Please be consistent in using a proper unit of “per milliliter”.
It was done (line 308).
Figure 4. Technically, superoxide anion is measured as “absorbance” to indicate formazan concentration and indicated as an absorbance unit. Therefore, it is not possible that the absorbance will be too high as shown in this figure.
The correction was done.
Line 299. Change “hemolymph” to “hemocytes”.
The correction was done (line 320).
3.3.3. Phenoloxidase activity in shrimp hemolymph
Figure 5. The unit of PO activity must be indicated as “unit enzyme/mL”. Please correct the graph and description in the text.
We only used the absorbance of the colorimetric reaction to determine the enzyme activity
Line 312. Italicize “L. vannamei”.
It was done.
3.4.1. Relative bacterial abundance in the shrimp gut
Line 331. Correct “bacteri”.
It was done.
Line 341. Italicize “Mesonia”.
It was done.
3.4.3. Beta diversity index
Line 377. Italicize “L. vannamei”.
It was done.
- Discussion
There are many errors and unnecessary references were cited. Most information of this section must be properly overhauled.
Line 415. The logic of the following content is wrong; “…as this organ is exposed to the water environment [45].”.
The correction was done (lines 422 and 423).
Line 433. Correct “[54] mention” to “Zhou et al. [54] mention” and the other places in the manuscript.
Ok.
Line 465. The following content is unnecessary; “[69] found that L. vannamei treated with 400 and 600 mg·L-1 of Gracilaria tenuistipitata extract for 3 h and then exposed to 5 mg·L-1 of ammonia, increased hemocytes compared to 466 the control group.”. The authors should step forward to describing the benefits of bacteria that properly reduce ammonia in culture water.
The citation and their information was deleted.
Similarly, the following contents must be removed.
1) Line 472-428. Similaly, [69] found that in the hemolymph of shrimp treated with G. tenuistipitata extract and then exposed to ammonia, the activity of the enzyme superoxide dismutase increased, which has the superoxide anion radical as a substrate. In contrast, [61] found no changes in SOD activity in the hepatopancreas of L. vannamei exposed to high nitrite concentrations, and [71] found similar results in L. vannamei pretreated with dietary inositol and exposed to ammonia during 24 h. Regarding phenoloxidase, its activity decreased in the treatments with LAB alone and the mixture of bacteria in the water. The decrease in phenoloxidase activity could indicate an optimal physiological state with less biotic stress. Converseley, data reported by [69] showed an increase in the activity of the enzyme in shrimp treated with G. tenuistipitata and ammonia.
Only part of this information was removed.
2) Line 486-490. In the work of [61], at the concentration of 2 mg L-1 of nitrites, the highest relative abundance corresponded to Bacteroidota, followed by Proteobacteria and Actinobacteria. At the highest nitrite concentration (20 mg L-1), the highest abundance corresponded to Proteobacteria followed by Bacteroidota and Actinobacteria.
This information was not eliminated since the authors mention the effect of nitrogenous waste on the intestinal microbiota.
Line 545 and 550. Correct grammatical errors of “which”.
Ok.
- Conclusion
As suggested in an earlier section, the term “nitrifying bacteria” should be avoided.
It has been shown that Bacillus licheniformis has the enzymes to be able to utilize TAN, nitrites and nitrates. According to the results, we can say that the bacilli and LAB acted as nitrifiers.
References
1) Please carefully correct inconsistent formats and other errors in some references. There are many errors in this part.
2) There are too many references used in this part. The authors should carefully reduce all necessary references in this part.
Some references have been removed and their formatting has been revised.
Reviewer 2 Report
Comments and Suggestions for Authors
Ms. No.: animals-3144748
Title: Investigating the effect of bacilli and lactic acid bacteria on water quality, growth, survival, immune response and intestinal microbiota of cultured Litopenaeus vannamei
Journal: Animals
This study aim to evaluate beneficial bacteria such as bacilli and lactic acid bacteria isolated from Litopenaeus vannamei and mixed with culture water with exchange, as obtained some positive results important to eliminate nitrogenous waste by bacteria. This Ms was found some important typho error as listed below clear before consider of this Ms.
Line 24: add ‘one of’ after Shrimp is
Line 27: TAN expand..
Line 28: LAB expand..
Line 30: which bacteria?
Line 40: add ‘identified’ after enteroides
Line 46: ‘market, generating’ change to ‘market and their production over’
Line 49: ‘add’ change to ‘supplement’
Line 49: add ‘which’ after equipment,
Line 52: ‘shrimp’ change to ‘culture organisms’
Line 57: ‘removal’ change to ‘elimination’
Line 62: ‘susceptible’ change to ‘vulnerable’
Line 64: remove ‘species such as’
Line 65: add ‘species’ after and Paracoccus
Line 77: (Intriago et al. 2018), change numerical number of the citation
Line 79: ‘[19])’ change to ‘[19]’
Line 92: Both Pediococcus pentosaceus and Leuconostoc mesenteroides were grown on MRS broth???
Line 97: All 3 bacterial were measure absorbance at 580 nm???
Line 98: How much concentration inoculated into the shrimp culture water???
Line 102: (NH4)2SO4, KNO3, MgSO4, KH2PO4, change to subscript of 2, 3, 4?
Line 118: Include water quality parameters here?
Line 127: All three Bacteria same 3 × 106 CFU·L-1 concentrations?
Line 127: placed in the water every 7 d, additionally add? Or after water exchange then add?, how determined the 3 × 106 CFU·L-1 concentrations?
Line 142: why (2:2:1) to obtain intestine samples?
Line 143:’ 1 mL’ change to ‘1-mL’
Line 154: EDTA-Na2, change 2 subscript?
Line 157, 162: hemolymph obtained from nine shrimp is pooled? Or individually determined THC?
Line 165, 174, 178, 180: space add ’37 °C’
Line 173-175: This sentence repeated, delete here
Line 181: 492 nm repeat in line 183, use any one place
Line 263, 268: (II), group II?
Line 264, 268: (IV), group IV?
Line 265, 295: (P > 0.05), P to be italics?
Line 277: water (II), LAB (III), and bacilli and LAB in the water (IV), use any one Group II, Group III and Group IV or water, LAB or bacilli + LAB, it is confusion?, change throughout the Ms..
Line 415: ‘very important’ change to ‘necessary’
Line 431: this is first report? If, yes, specify?
Line 433: Add ‘Zhou et al.’ before ‘[54]’
Line 440: add ‘(SCFAs)’ after ‘short-chain fatty acids’
Line 447: ‘[64, 65]).) [66]’ change to ‘[64, 65]. Boardman et al. [66]’
Line 466: ‘increased hemocytes’ change to ‘raised hemocytes counts’
Line 469: Add ‘Liao et al.’ before [70]
Line 479: mixture of which bacteria???
Comments on the Quality of English Language
Ms. No.: animals-3144748
Title: Investigating the effect of bacilli and lactic acid bacteria on water quality, growth, survival, immune response and intestinal microbiota of cultured Litopenaeus vannamei
Journal: Animals
This study aim to evaluate beneficial bacteria such as bacilli and lactic acid bacteria isolated from Litopenaeus vannamei and mixed with culture water with exchange, as obtained some positive results important to eliminate nitrogenous waste by bacteria. This Ms was found some important typho error as listed below clear before consider of this Ms.
Line 24: add ‘one of’ after Shrimp is
Line 27: TAN expand..
Line 28: LAB expand..
Line 30: which bacteria?
Line 40: add ‘identified’ after enteroides
Line 46: ‘market, generating’ change to ‘market and their production over’
Line 49: ‘add’ change to ‘supplement’
Line 49: add ‘which’ after equipment,
Line 52: ‘shrimp’ change to ‘culture organisms’
Line 57: ‘removal’ change to ‘elimination’
Line 62: ‘susceptible’ change to ‘vulnerable’
Line 64: remove ‘species such as’
Line 65: add ‘species’ after and Paracoccus
Line 77: (Intriago et al. 2018), change numerical number of the citation
Line 79: ‘[19])’ change to ‘[19]’
Line 92: Both Pediococcus pentosaceus and Leuconostoc mesenteroides were grown on MRS broth???
Line 97: All 3 bacterial were measure absorbance at 580 nm???
Line 98: How much concentration inoculated into the shrimp culture water???
Line 102: (NH4)2SO4, KNO3, MgSO4, KH2PO4, change to subscript of 2, 3, 4?
Line 118: Include water quality parameters here?
Line 127: All three Bacteria same 3 × 106 CFU·L-1 concentrations?
Line 127: placed in the water every 7 d, additionally add? Or after water exchange then add?, how determined the 3 × 106 CFU·L-1 concentrations?
Line 142: why (2:2:1) to obtain intestine samples?
Line 143:’ 1 mL’ change to ‘1-mL’
Line 154: EDTA-Na2, change 2 subscript?
Line 157, 162: hemolymph obtained from nine shrimp is pooled? Or individually determined THC?
Line 165, 174, 178, 180: space add ’37 °C’
Line 173-175: This sentence repeated, delete here
Line 181: 492 nm repeat in line 183, use any one place
Line 263, 268: (II), group II?
Line 264, 268: (IV), group IV?
Line 265, 295: (P > 0.05), P to be italics?
Line 277: water (II), LAB (III), and bacilli and LAB in the water (IV), use any one Group II, Group III and Group IV or water, LAB or bacilli + LAB, it is confusion?, change throughout the Ms..
Line 415: ‘very important’ change to ‘necessary’
Line 431: this is first report? If, yes, specify?
Line 433: Add ‘Zhou et al.’ before ‘[54]’
Line 440: add ‘(SCFAs)’ after ‘short-chain fatty acids’
Line 447: ‘[64, 65]).) [66]’ change to ‘[64, 65]. Boardman et al. [66]’
Line 466: ‘increased hemocytes’ change to ‘raised hemocytes counts’
Line 469: Add ‘Liao et al.’ before [70]
Line 479: mixture of which bacteria???
Author Response
Reviewer 2
Line 24: add ‘one of’ after Shrimp is
It was done.
Line 27: TAN expand..
It was done.
Line 28: LAB expand..
Line 30: which bacteria?
- licheniformis, Pediococcus pentosaceus and Leuconostoc mesenteroides.
Line 40: add ‘identified’ after enteroides
It was done.
Line 46: ‘market, generating’ change to ‘market and their production over’
It was done.
Line 49: ‘add’ change to ‘supplement’
Ok.
Line 49: add ‘which’ after equipment,
Ok.
Line 52: ‘shrimp’ change to ‘culture organisms’
Ok
Line 57: ‘removal’ change to ‘elimination’
Ok.
Line 62: ‘susceptible’ change to ‘vulnerable’
Ok.
Line 64: remove ‘species such as’
Ok
Line 65: add ‘species’ after and Paracoccus
It was done (Line 66).
Line 77: (Intriago et al. 2018), change numerical number of the citation
Ok.
Line 79: ‘[19])’ change to ‘[19]’
Ok
Line 92: Both Pediococcus pentosaceus and Leuconostoc mesenteroides were grown on MRS broth???
Yes (Lines 95-97).
Line 97: All 3 bacterial were measure absorbance at 580 nm???
It is correct (Line 100).
Line 98: How much concentration inoculated into the shrimp culture water???
Each bacteria (3 × 106 CFU·L-1) were placed in the water every 7 d (Lines 130,131).
Line 102: (NH4)2SO4, KNO3, MgSO4, KH2PO4, change to subscript of 2, 3, 4?
It was done.
Line 118: Include water quality parameters here?
Water quality parameters were determined only in the bioassay.
Line 127: All three Bacteria same 3 × 106 CFU·L-1 concentrations?
Each one.
Line 127: placed in the water every 7 d, additionally add? Or after water exchange then add?, how determined the 3 × 106 CFU·L-1 concentrations?
There was no water Exchange. The bacteria were previously counted in the aforementioned Works (Line 92).
Line 142: why (2:2:1) to obtain intestine samples?
Yes (Line 146).
Line 143:’ 1 mL’ change to ‘1-mL’
Ok.
Line 154: EDTA-Na2, change 2 subscript?
Ok.
Line 157, 162: hemolymph obtained from nine shrimp is pooled? Or individually determined THC?
Individual extraction (157).
Line 165, 174, 178, 180: space add ’37 °C’
Ok.
Line 173-175: This sentence repeated, delete here
It was done.
Line 181: 492 nm repeat in line 183, use any one place.
The correction was done.
Line 263, 268: (II), group II?
Yes.
Line 264, 268: (IV), group IV?
Yes.
Line 265, 295: (P > 0.05), P to be italics?
Ok.
Line 277: water (II), LAB (III), and bacilli and LAB in the water (IV), use any one Group II, Group III and Group IV or water, LAB or bacilli + LAB, it is confusion?, change throughout the Ms..
Throughout the document, the control and treatments with the bacterial species inoculated into the water individually and the mixture of the three bacteria (treatment IV) are mentioned.
Line 415: ‘very important’ change to ‘necessary’
Ok.
Line 431: this is first report? If, yes, specify?
We put: there are no reports indicating Pediococcus and Leuconostoc as nitrifying bacteria (Lines 438,439).
Line 433: Add ‘Zhou et al.’ before ‘[54]’
Ok.
Line 440: add ‘(SCFAs)’ after ‘short-chain fatty acids’
The correction was done.
Line 447: ‘[64, 65]).) [66]’ change to ‘[64, 65]. Boardman et al. [66]’
Ok.
Line 466: ‘increased hemocytes’ change to ‘raised hemocytes counts’
Ok.
Line 469: Add ‘Liao et al.’ before [70]
Ok.
Line 479: mixture of which bacteria???
We put “bacilli and LAB” (Line 484).
Round 2
Reviewer 1 Report
Comments and Suggestions for Authors
Reviewers' Comments to Authors:
The revised manuscript entitled “Investigating the Effect of Bacilli and Lactic Acid Bacteria on Water Quality, Growth, Survival, Immune Response and Intestinal Microbiota of Cultured Litopenaeus vannamei” by Vega-Carranza et al. demonstrates the scientific attempts to explain the efficacy of probiotic application on growth, immune response and gut microbiota of Pacific white shrimp.
The manuscript has been significantly improved in the current version, and most recommendations raised previously have been properly addressed. However, it still contains some errors that the authors must pay more attention to address and improve the quality of this manuscript to meet an acceptable high-standard journal.
The recommendation of “accepting” could be considered after correcting the following minor concerns.
Simple Summary
Line 21. Correct “Bacilli” to “bacilli”.
Abstract
Line 28-29. The following content should not be redundant with the content in “Simple Summary”.
1. Introduction
Line 53. Correct a grammatical error of “…is generated, which…”.
Line 74. Clarify “bound to hemocytes”, this is a piece of wrong information.
2. Materials and Methods
Line 127. Clarify “(600 shrimp m^3).
Line 179-180. Correct the following information; “The optical density of the dissolved formazan was measured at 630 nm (OD630)” to “The absorbance of the dissolved formazan was read at the optical density (OD) at 630 nm”.
Line 192-193. After absorbance reading, the authors must calculate the “unit enzyme/mL” to express the concentrations of detected enzymes, which will represent PO activity.
Line 240. Correct “FO activity” to “PO activity”.
3. Results
Figure 1. The information is shown in this part. The authors should provide and clearly indicate “Figure 1A, B, C”.
Figure 5. The unit of PO activity must be indicated as “unit enzyme/mL”. Please correct the graph and description in the text.
Table 3. Please correct the wrong information found in “ACE” columns I and II.
Please keep consistently using “p (Italic form)” instead “P” throughout the manuscript.
References
1) Please carefully correct inconsistent formats and other errors in some references. There are some errors in this part such as references; 5, 30, 36, 40,45, 48, 49, 64, 76, 81, 90, 95 and 99.
Comments on the Quality of English Language-
Author Response
Reviewer 1
Simple Summary
Line 21. Correct “Bacilli” to “bacilli”.
Okay.
Abstract
Line 28-29. The following content should not be redundant with the content in “Simple Summary”.
The correction was done.
- Introduction
Line 53. Correct a grammatical error of “…is generated, which…”.
Okay.
Line 74. Clarify “bound to hemocytes”, this is a piece of wrong information.
It was deleted. We put only “plasma”.
- Materials and Methods
Line 127. Clarify “(600 shrimp m^3).
We put “equivalent to 600 shrimp m3.
Line 179-180. Correct the following information; “The optical density of the dissolved formazan was measured at 630 nm (OD630)” to “The absorbance of the dissolved formazan was read at the optical density (OD) at 630 nm”.
The correction was done.
Line 192-193. After absorbance reading, the authors must calculate the “unit enzyme/mL” to express the concentrations of detected enzymes, which will represent PO activity.
The change was done.
Line 240. Correct “FO activity” to “PO activity”.
Ok.
- Results
Figure 1. The information is shown in this part. The authors should provide and clearly indicate “Figure 1A, B, C”.
The correction was done.
Figure 5. The unit of PO activity must be indicated as “unit enzyme/mL”. Please correct the graph and description in the text.
Ok.
Table 3. Please correct the wrong information found in “ACE” columns I and II.
Okay.
Please keep consistently using “p (Italic form)” instead “P” throughout the manuscript.
Okay.
References
1) Please carefully correct inconsistent formats and other errors in some references. There are some errors in this part such as references; 5, 30, 36, 40,45, 48, 49, 64, 76, 81, 90, 95 and 99.
We checked all references and Bradford reference was include (number 28).
Reviewer 2 Report
Comments and Suggestions for Authors
Article ref. no.: animals-3144748
Title: Investigating the Effect of Bacilli and Lactic Acid Bacteria on Water Quality, Growth, Survival, Immune Response and Intestinal Microbiota of Cultured Litopenaeus vannamei
Journal: Animals
Line 33: All three bacteria?
Line 37: lower pathogenic or beneficial bacterial species richness?
Line 93: Bacillus licheniformis BCR 4-3 isolated from which source? Mention here
Line 95: Pediococcus pentosaceus isolated from which source? Mention here
Line 96: Leuconostoc mesenteroides isolated from which source? Mention here
Line 119: ‘animals’ change to ‘shrimp’
Line 125: Feeding experiment started after 15 d acclimatized period, during the 15 d, no growth was observed???, as mention in line 114 and 125
Line 126: plastic tanks measurement to be include here
Line 131: How can calculate and determine and adjust in each bacteria at 3 × 106 CFU·L-1 concentration???
Line 133: include ‘during the experimental period’ after exchange
Line 171: three shrimp per tank (nine per treatment) were centrifuged, all 9 samples were pooled and centrifuged? Or individually centrifuged?
Figure 1, is not visible clearly (Line 267-276).
Figure 6, 7, 8, 9, not visible clearly
Line 475: ‘organisms’ change to ‘shrimps’
Comments on the Quality of English LanguageArticle ref. no.: animals-3144748
Title: Investigating the Effect of Bacilli and Lactic Acid Bacteria on Water Quality, Growth, Survival, Immune Response and Intestinal Microbiota of Cultured Litopenaeus vannamei
Journal: Animals
Line 33: All three bacteria?
Line 37: lower pathogenic or beneficial bacterial species richness?
Line 93: Bacillus licheniformis BCR 4-3 isolated from which source? Mention here
Line 95: Pediococcus pentosaceus isolated from which source? Mention here
Line 96: Leuconostoc mesenteroides isolated from which source? Mention here
Line 119: ‘animals’ change to ‘shrimp’
Line 125: Feeding experiment started after 15 d acclimatized period, during the 15 d, no growth was observed???, as mention in line 114 and 125
Line 126: plastic tanks measurement to be include here
Line 131: How can calculate and determine and adjust in each bacteria at 3 × 106 CFU·L-1 concentration???
Line 133: include ‘during the experimental period’ after exchange
Line 171: three shrimp per tank (nine per treatment) were centrifuged, all 9 samples were pooled and centrifuged? Or individually centrifuged?
Figure 1, is not visible clearly (Line 267-276).
Figure 6, 7, 8, 9, not visible clearly
Line 475: ‘organisms’ change to ‘shrimps’
Author Response
Reviewer 2
Line 33: All three bacteria?
The correction was done.
Line 37: lower pathogenic or beneficial bacterial species richness?
The analysis does not specify that.
Line 93: Bacillus licheniformis BCR 4-3 isolated from which source? Mention here
Okay.
Line 95: Pediococcus pentosaceus isolated from which source? Mention here
Okay.
Line 96: Leuconostoc mesenteroides isolated from which source? Mention here
Okay.
Line 119: ‘animals’ change to ‘shrimp’
Okay.
Line 125: Feeding experiment started after 15 d acclimatized period, during the 15 d, no growth was observed???, as mention in line 114 and 125
The weight of the collected shrimp was recorded before acclimatization.
Line 126: plastic tanks measurement to be include here
Okay.
Line 131: How can calculate and determine and adjust in each bacteria at 3 × 106 CFU·L-1 concentration???
Please, see line 103.
Line 133: include ‘during the experimental period’ after Exchange
Okay.
Line 171: three shrimp per tank (nine per treatment) were centrifuged, all 9 samples were pooled and centrifuged? Or individually centrifuged?
Each one. The correction was done.
Figure 1, is not visible clearly (Line 267-276).
Figure 6, 7, 8, 9, not visible clearly
We try to improve the figures.
Line 475: ‘organisms’ change to ‘shrimps’
Okay.